# Laboratory Analyses Used to Define the Nutritional Parameters and Quality Indexes of Some Unusual Forages

**DOI:** 10.3390/ani12182320

**Published:** 2022-09-07

**Authors:** Sonia Tassone, Sabah Mabrouki, Salvatore Barbera, Sara Glorio Patrucco

**Affiliations:** Department of Agriculture, Forestry, and Food Sciences, University of Turin, 10095 Grugliasco, Italy

**Keywords:** unusual forages, laboratory analyses, measured and calculated parameters, quality indexes

## Abstract

**Simple Summary:**

In the present study, laboratory analyses conducted on unusual forages have been used to calculate various parameters and define some quality indexes using different equations. It was found that the quality of unusual forages decreased during the growth but maintained a high level. Different indexes were used to evaluate the forage quality. The Relative Feed Value and the Relative Forage Quality indexes, when calculated with the formula for legumes, correctly summarized the trend of the nutritional characteristics of the unusual forages at different maturity stages.

**Abstract:**

The quality of a forage influences the production of animals, and it can be defined in many ways. Laboratory analyses are important tools because they can be used to indicate the quality of the forages, and they represent a relatively quick way of defining their nutritive values. However, specific quality indexes are necessary to evaluate and rank forages. The quality of conventional forages is predicted by different indexes, according to whether they are legumes or grasses. However, no indications are given about what formulae should be used for unusual forages. In the present study, laboratory analyses have been conducted on three unusual crops belonging to three different botanical families (amaranth, borage, and camelina) at four growth stages, and conventional quality indexes have been calculated and applied to establish their quality. The obtained results have shown that the nutritive value of the unusual forages modified during the growth, although they always maintained a high quality. Hence, the Relative Feed Value of unusual forages can be measured using the ADF content or digestibility value. The Relative Forage Quality, calculated with the legume formula, seems more appropriate for the considered unusual forages as it was able to reveal any changes that took place during maturity.

## 1. Introduction

Laboratory analyses are very important to establish the nutritive value of feeds and to quantify the presence and the availability of nutrients that influence their voluntary intake and animal production [1]. The terms “forage nutritive value” and “forage quality” are generally used interchangeably. However, “forage nutritive value” just refers to the concentrations of available energy and crude protein, while “forage quality” is associated with energy, proteins, fibers, minerals, vitamins, digestibility, and also forage intake [2].

Forage quality can be defined by means of numerous parameters, which are either measured in the laboratory, through chemical and physical analyses, or are calculated using the determined parameters. One problem that arises when defining forage quality is the need to reconcile the various levels of nutrients reported by means of a laboratory analysis and then to evaluate and rank the forages according to their quality. One solution to this problem is to calculate indexes that provide a single number that can be used to differentiate the quality of forages.

Over the course of time, several indexes, and parameters, based on analytical data, have been defined and developed to estimate forage quality [3]. For example, the Relative Feed Value (RFV) index represents the digestibility and intake potential (DMI) of a forage, as predicted from the ADF and NDF contents, respectively [2]. However, the DMI has been underestimated in higher-quality grasses [3,4,5], as has the RFV [6]. For these reasons, the RFV index, which has been reported in documents and textbooks [7], is used for the marketing of forages, albeit just for temperate species, but it is not used in nutritional models.

With the advent of the dynamic computer model to describe biological systems, different feed evaluation methods have been developed for cattle [8,9]. The Italian National Research Council [10] has introduced a new and dynamic approach to estimate the energy value of feeds, diets, and their intake in lactating cows, which uses new analyses and equations, thereby improving the generally used quality indexes [11]. This approach uses the available energy of feeds, calculated as the total digestible nutrients (TDN) and the DMI. The TDN is obtained from the nutrient composition of a forage, while the protein, crude fiber, nitrogen free extract, and fat (each digestible nutrient) are summed and multiplied by their true digestibility. Such a value is expressed as net energy of lactation units (NE_L_) and varies according to the level of maturity (TDN decrease) and the type of forage (Alfalfa > cool-season grasses > clovers > warm-season grasses) [2]. The dry matter intake of lactating cows is predicted by means of a universal equation that is applicable for all lactations and stages of lactation, considering that the intake is closely related to the NDF content [1]. Weiss et al. [12] tested the accuracy of this approach by calculating the energy intake and verified that the system is accurate. Moreover, if a diet has a restricted number of forages, the TDN on its own can offer indications on the quality of forages [13]. When a forage is fed on its own, the TDN intake can be used to predict the associative effect between forages and concentrates [14]. The TDN and DMI allow the Relative Forage Quality (RFQ) to be calculated. The RFQ has been proposed as an alternative to the RFV, as it is more appropriate for the prediction of forage quality. It considers the digestible fiber (NDFD) in a diet and can be applied to all forages, except corn silage. It is calculated with specific equations for different types of forage (legume or grass) [2].

Quality indexes can be used for traditional and unusual forages [11,15]. Different equations are used to calculate the same index, starting from different analytical parameters. Although traditional forages belong to a specific category (legumes or seasonal grasses), unusual forages are often from other families, and it is therefore difficult to choose an appropriate equation. Unusual forages are principally cultivated for their seeds, which are used in human nutrition, but also for their stems and leaves, which can also be used as a forage [16,17,18,19].

In this study, laboratory analyses have been used to obtain and calculate the nutritional parameters of unusual forages at different maturity stages and to calculate some indexes as indicators of their quality.

## 2. Materials and Methods

### 2.1. Plant Material

Three sets of data, pertaining to amaranth, borage, and camelina, which were determined in previous studies (Table 1), have been used in this experiment. Overall, the unusual forage samples were analyzed to define the chemical composition and digestibility of the dry matter and neutral detergent fiber [16,17,18]. Then, on the basis of the results, other nutritional parameters and quality indexes were calculated. The analytical data were grouped into four common stages of maturity, vegetative (1), elongation (2), reproductive (3), and seed (4) [20], to allow comparisons to be made.

### 2.2. Measured Parameters

The results of the chemical analyses were collected from the database of previous articles by Peiretti et al. [16,17,18]. Because we used data that had been published in previous works to calculate the other parameters and the indexes, the other data that were necessary for such a calculation, but which were not available, were specifically determined.

Digestibility was measured *in vitro* using Ankom Daisy^II^ Technology (Ankom Technology Corporation, Fairport, NY, USA), while cattle rumen fluid was uesd as *inoculum*. Dry matter and neutral detergent fiber digestibility (ADMD_AD_^II^ and NDFD_AD_^II^, respectively [21]) were calculated as reported by Tassone et al., 2020b [22]:

ADMD_AD_^II^ (g kg^−1^ DM) = 1000 × (DM0h − DMresidue)/DM0h

NDFD_AD_^II^ (g kg^−1^ NDF) = 1000 × (NDF0h − NDFresidue)/NDF0h

where:

DM0h (g kg^−1^) = dry matter *ante incubation;*

DMresidue (g kg^−1^) = dry matter *post incubation;*

NDF0h (g/kg^−1^ DM) = neutral detergent fiber *ante incubation;*

NDFresidue (g/kg^−1^ NDF) = neutral detergent fiber *post incubation*.

### 2.3. Calculated Parameters

Some parameters were calculated on the basis of laboratory results to avoid the necessity of conducting other analyses [11]. The calculated parameters were non-fibrous carbohydrate (NFC) [10,11], fatty acids (FA), nitrogen-free NDF (NDFn) [11], and dry matter digestibility (ADF based) [23]. Moreover, by using both equations for legumes (lg) and grasses (gr), the total digestible nutrients, (TDN_lg_ and TDN_gr_ for legumes and grasses, respectively), the dry matter intake on a TDN basis (DMI_lg_, DMI_gr_) [24,25] and on an NDF basis (DMI_NDF_) [23], and the net energy of lactation, calculated on an ADF basis [10,26], were obtained, as described in Table 2.

### 2.4. Calculated Indexes

The parameters determined by means of laboratory analyses and those calculated with the formulae that resorted to analytical data were used in equations to calculate some quality indexes [13], as shown in Table 2.

The RFV index was calculated using dry matter digestibility values measured *in vitro* (RFV_vt_) and calculated using the ADF content (RFV_ADF_).

The RFQ index was estimated using equations for both a legume and a grass mixture (RFQ_lg_) and for warm- and cool-season grasses (RFQ_gr_).

### 2.5. Statistical Analyses

The data were grouped into four common stages of maturity and statistically analyzed by means of a one-way variance analysis using PROC GLM of SAS (Statistical Analysis System Institute, Cary, NC, USA, 2021) [27] to compare the stages of maturity within each species. The Tukey test for multiple Lsmean comparisons was performed. For each combination of species and stage of maturity, there were at least three replicates within the same year, and the year factor was not considered as it was not always present.

## 3. Results

The mean values of the measured parameters of each species, expressed on a dry matter (DM) basis, for different stages of maturity, which were necessary to calculate the indexes, are reported in Table 3. The dry matter content generally increased during growth [16]. However, the borage maintained very low DM values, similar between the vegetative and reproductive stages (Table 3).

The ash content of the amaranth had higher values at the vegetative stage but maintained similar values throughout the growth, while the values were higher for the borage and only decreased at the seed stage. In the camelina, it decreased during the growth. The CP showed higher values at the vegetative stage for all the species (Table 3) and maintained similar values during the elongation, reproductive, and seed stages. The NDF varies significantly from the vegetative to the other stages in the amaranth, unlike the camelina, which increased considerably after the first stage. The borage maintained low and similar NDF values after the vegetative stage. The ADF content in the amaranth and borage increased significantly after the vegetative stage, maintaining similar values during the growth, while the lowest value in the camelina was observed at the vegetative stage, increased in the elongation stage, and it was similar in the other stages. The EE extract content did not change during the growth of the amaranth, while in the borage, we found higher values just during the vegetative stage. The camelina showed a high EE extract content that decreased between the vegetative and reproductive stages. The ADMD_AD_^II^ of all the plants was high and decreased with their maturity. A similar trend with high values was observed for the NDFD_AD_^II^ (Table 3), particularly for the amaranth during the vegetative stage, but with a high variability. In the borage, it decreased after the elongation stage, and in the camelina, it maintained similar values between the elongation and the reproductive stages. The NE_L_ was significantly higher at the vegetative stage and decreased immediately for the borage and camelina, while it decreased after the reproductive stage for the amaranth.

The parameters that were calculated using the chemical analysis values are reported in Table 4. The NFC decreased after the reproductive stage in the amaranth and after the vegetative stage in the camelina. In the borage, it increased during the growth. The FA values were similar throughout the growth for the amaranth, while they decreased after the elongation stage for the borage and after the vegetative stage for the camelina. The NDFn had similar values at the vegetative and the reproductive stages for the amaranth, while in the borage and camelina, it increased after the vegetative stage, albeit just at the reproductive stage. The dry matter digestibility of the amaranth and borage, calculated on an ADF basis, decreased after the vegetative stage, and it was similar in the later stages. In the camelina, it decreased between the vegetative and reproductive stages. The DMI_NDF_ was higher for all the species in the vegetative stage, then decreased, and it was similar in the later stages just for the amaranth. The same trend was observed when using the DMI calculated with the legume formula, while significant differences were observed when using the grass formula just for the camelina. The TDN_lg_ was higher up to the reproductive stage for the amaranth, while in the camelina, it decreased after the vegetative stage. An average TDN_lg_ value of 552 g kg^−1^ DM was observed for the borage (Table 4), without any significant differences being observed during maturity. The TDN_gr_ decreased after the vegetative stage, albeit just in the camelina. The NE_L_ was higher just in the vegetative stage for the amaranth and borage, while it decreased up to the elongation stage in the camelina.

The RFV and RFQ values, calculated considering different chemical characteristics (the ADF and *in vitro* digestibility for the RFV) and the equations for legumes and grasses (RFQ_lg_ and RFQ_gr_), respectively, are reported in Table 5. In general, the RFV_ADF_ showed lower values than the RFV, when calculated using forage digestibility, and the RFQ_lg_ was higher than the RFQ_gr_. Regardless of which index was used, the amaranth showed the best indexes at the vegetative stage, and no important qualitative differences were observed during maturity. The RFQ_gr_ of the amaranth was similar for all the stages. The borage presented the same indexes for all the stages, and only the RFV_ADF_ detected a decrease at the seeding stage and the RFV_vt_ at the vegetative stage. The camelina presented a significantly decrease in all the indexes during the growth.

## 4. Discussion

Information on forage quality should be made available to farmers before feeding. The use of laboratory analyses on forages allows their chemical composition and digestibility to be defined and the intake and total digestible nutrients that have to be used to predict forage quality to be calculated [2]. Such analyses have long been used for conventional forages to develop indexes and to summarize their quality in a value [28]. However, very little information about the nutritional characteristics of unusual forages is available and, consequently, no specific quality index has been developed. In this paper, measured and calculated nutritional values are reported for three unusual forages: amaranth, borage, and camelina. Moreover, indexes used for traditional forages have been applied to unusual forages using a different formula.

Considering the results of the laboratory analyses, it can be observed, in the same way as for conventional forages [29,30], that the development stage also influenced the chemical composition (Table 3) in the unusual forages [16,17,18]. An increase in dry matter was particularly consistent for the camelina. The CP values were high, even when compared with other legumes and grass forages [31,32]. In particular, the CP content was high at the vegetative stage for all the species (average of 202 g kg^−1^) (Table 3), considering that *Medicago sativa* ranged between 225 and 246 g kg^−1^ DM when referring to the same stage [33]. Unlike *Medicago sativa,* with its 178 g kg^−1^ DM during flowering [33], the protein content in the unusual forages was significantly reduced after the elongation stage (up to 100 g kg^−1^ DM). The NDF and ADF increased with the maturity of the plant (Table 3). Generally, they were similar to those of other vegetable types and species in other studies [32,34] or, on average, lower than others [1,31,35]. The obtained results showed that the fiber fractions for the amaranth and borage were similar from the elongation stage up to the seed stage. The ADF content of all the species influenced the DM digestibility. The ADMD_AD_^II^ showed higher values than the other species [36], with a trend that decreased significantly after the vegetative stage in the amaranth and camelina. The borage maintained high digestibility during the growth. Similarly, the NDFD_AD_^II^ was very high at the vegetative stage, and in particular for the amaranth (Table 3). The average values were similar to those found for grass and legume winter crops [32] and were higher for the amaranth and borage than tropical species [1]. The lower values of the camelina were an indication of an increase in its fibrous fractions during the growth.

The parameters measured in the laboratory allowed other important parameters to be calculated in order to define the quality of the unusual forage (Table 4). For example, non-fibrous carbohydrates represent a readily available portion of feeds and, as such, positively reflect on the evaluation of the feed quality. The values higher than 212 g kg^−1^ DM found in the amaranth, borage, and camelina indicated a high quality, especially when associated with low fiber values [37]. Other parameters, such as the FA, NDFn, and ADMD_ADF,_ were calculated to avoid the need for conducting specific analyses, and thus to reduce costs, times, and equipment. If we compare the digestibility values obtained in vitro with the calculated ones, we can observe that the ADMD_ADF_ showed lower values and less variables. The borage was found to be highly digestible (Table 4). Compared with conventional forages, its digestibility was similar to that of legumes, as the DM digestibility is usually lower in warm-season forages (440–660 g kg^−1^), intermediate to high in cool-season forages (490–810 g kg^−1^), and higher in legumes (690–810 g kg^-^) [38]. The DMI represents an important parameter, especially for forages, as it depends to a great extent on the fiber concentration. The fiber increased during maturity and, consequently, the DMI values decreased significantly after the vegetative stage when it was calculated using the NDF content, and with the formula for legumes (Table 4). No significant differences were observed when using the grass formula for the amaranth and borage. The DMI_NDF,_ DMI_lg_, and DMI_gr_ showed higher values (Table 4) than the values found for the other species [1,13,35]. The TDN reported the quantity of digestible material in the forage, and when calculated with the legume formula, the values decreased significantly from the vegetative stage for the camelina (Table 4). The average values were higher in all the cases than for other species [1,13,35]. The energy value of the forage, expressed as NE_L,_ decreased together with the forage nutritive value and showed a similar trend and similar values for the amaranth and borage.

An accurate prediction of the described parameters improves the precision of the quality indexes. The RFV calculated using the ADF content resulted in lower values than when it was calculated with the digestibility values, although the trend was similar for the amaranth and camelina (Table 5). However, all the unusual forages that have been analyzed here could be considered as high-quality forages [39,40].

The RFQ_lg_ showed a decrease in the quality for the amaranth and camelina during maturity from the vegetative stage, while the RFQ_gr_ showed significantly decreased values, albeit just for the camelina, during the growth. The RFQ_lg_ values were higher than the RFQ_gr_ ones but were higher than other legumes, grasses, and tropical species. As indicated in Table 5, the RFQ_gr_ was not able to detect any changes in the nutritional composition of the amaranth or borage during maturity.

Summarizing, the indexes that better represented the forage quality seem to be, respectively, the RFV_ADF_, RFV_vt_, and RFQ_lg_ for amaranth and the RFV_ADF_ and RFV_vt_ for borage; for camelina, all the indexes could be good.

## 5. Conclusions

Forage analyses are important because they indicate the quality of a forage, and they allow the nutritive value of the forage that will be grazed and the hay that will be purchased or placed on the market to be evaluated [2].

The species analyzed here are of great interest for researchers as they represent promising plant resources, can have a high productivity, and are able to adapt to any growing conditions. Moreover, they represent a source of protein, with nutraceutical potentiality (bioactive compounds, essential fatty acids, tocopherol, squalene, and phenolic contents), and could also be used as by-products [41,42,43,44].

The amaranth, borage, and camelina considered in our study showed a good quality level, in particular at the vegetative stage, which decreased over time, but maintained high values up to the seed stage. However, we should also consider the presence of anti-quality factors, which is one of the main problems with alternative feeds [44], including phytic acids, tannins, oxalates, enzyme inhibitors, saponins, and nitrates, in particular for the amaranth [44], and glucosinolates and sinapine for camelina [45]. These substances diminish the nutritional requirements that are necessary to satisfy a specific kind and class of animal, although several processing methods are available that can be applied to reduce them [44].

The accuracy of quality indexes is a function of the accuracy of the laboratory analyses.

In this work, it has been shown that some quality indexes used for conventional forages can also be used to predict the quality of unusual forages. In particular, the indexes that were more able to represent the trend of nutritive characteristics during maturity were: the RFV, calculated using the ADF content or digestibility value, and the RFQ, calculated with the legume formula (RFQ_lg_). The RFQ calculated with the grass formula (RFQ_lg_) was not so successful for all the species analyzed, as it did not detect the nutritive changes during the maturity of the considered unusual forages.

## Figures and Tables

**Table 1 animals-12-02320-t001:** Species, botanic family, and reference.

Forage	Species	Botanic Family	Reference
Amaranth	*Amaranthus caudatus*	*Amarantaceae*	[16]
Borage	*Borago officinalis*	*Boraginaceae*	[17]
Camelina	*Camelina sativa*	*Brassicaceae*	[18]

**Table 2 animals-12-02320-t002:** Parameters and indexes calculated to estimate forage quality.

Parameter/Index	Acronym	Formula	Reference
Non-Fibrous Carbohydrate − g kg^−1^ DM	NFC	NFC = 100 − (NDFn + CP + EE + ash)	[10,11]
Fatty Acids − g kg^−1^ DM	FA	FA = EE − 1	[11]
Nitrogen-free NDF g kg^−1^ DM	NDFn	NDFn = NDF × 0.93	[11]
Dry Matter Digestibility g kg^−1^ DM (ADF based)	DMD_ADF_	DMD_ADF_ = 88.9 − (0.779 × ADF)	[23] [13]
Total Digestible Nutrient g kg^−1^ DM	TDN	TDN_lg_ = (NFC × 0.98) + (CP × 0.93) + (FA × 0.97 × 2.25) + (NDFn × (NDFD/100) − 7 TDN_gr_ = (NFC × 0.98) + (CP × 0.87) + (FA × 0.97 × 2.25) + (NDFn × (NDFD/100) − 10	[10]
Dry Matter Intake (NDF based) % BW	DMI_NDF_	DMI_NDF_ = 1.2/(NDF × 0.01)	[23]
Dry Matter Intake % BW	DMI	DMI_lg_ = 120/NDF + (NDFD − 45) × 0.374/1350 × 100 DMI_gr_ = −2.318 + 0.442 × CP − 0.0100 × CP^2^ − 0.0638 × TDN_gr_ + 0.000922 × TDN_gr^2^_ + 0.180 × ADF − 0.00196 × ADF^2^ − 0.00529 × CP × ADF	[24,25]
Relative Feed Value	RFV	RFV_vt_ = (DMI_NDF_ × DMD_vt_)/1.29	
		RFV_ADF_= (DMI_NDF_ × DMD_ADF_)/1.29	
Relative Forage Quality	RFQ	RFQ_lg_ = (DMI_lg_ × TDN_lg_)/1.23 RFQ_gr_ = (DMI_gr_ × TDN_gr_)/1.23	[12]
Net Energy of Lactation Mcal kg^−1^ (ADF basis)	NE_L_ADF_	NE_L_ADF_ = (0.866 − (0.0077 × ADF)) × 2.2	[10,26]

NDF_n_ = neutral detergent fiber nitrogen free; CP = crude protein; EE = ether extract; NDF = neutral detergent fiber; ADF = acid detergent fiber; _lg_ = legumes; _gr_ = grass; NDFD = neutral detergent fiber digestibility; TDN_lg_ = total digestible nutrient for legumes; TDN_gr_ = total digestible nutrient for grasses; _vt_ = in vitro; DMD_vt_ = dry matter digestibility measured in vitro; DMD_ADF_ = dry matter digestibility—ADF based; g/kg DM = g/kg of dry matter; % BW = percentage of body weight.

**Table 3 animals-12-02320-t003:** Lsmean and mean square of error (MSE) for the measured parameters at the vegetative, elongation, reproductive, and seed stages (Amaranth DFE = 10; Borage DFE = 6; Camelina DFE = 11).

Parameter	Unit	Species	Stage of Maturity				MSE
			Vegetative	Elongation	Reproductive	Seed	
DM	g kg^−1^ DM	Amaranth	119 ^C^	142 ^B^	179 ^A^	160 ^AB^	58.7
		Borage	78 ^b^	78 ^b^	84 ^b^	99 ^a^	26
		Camelina	112 ^C^	176 ^B^	194 ^B^	254 ^A^	283
Ash	g kg^−1^ DM	Amaranth	210 ^a^	163 ^b^	148 ^b^	153 ^b^	434
		Borage	261 ^a^	249 ^ab^	214 ^ab^	194 ^b^	354
		Camelina	139 ^A^	106 ^B^	85 ^C^	67 ^D^	40
CP	g kg^−1^ DM	Amaranth	187 ^a^	102 ^b^	73 ^b^	106 ^ab^	1482
		Borage	199 ^aA^	155 ^bAB^	142 ^B^	126 ^B^	83
		Camelina	220 ^A^	147 ^B^	124 ^aBC^	100 ^bC^	105
NDF	g kg^−1^ DM	Amaranth	360 ^cB^	444 ^bA^	427 ^bAB^	515 ^aA^	772
		Borage	288 ^C^	324 ^bAB^	320 ^B^	344 ^aA^	18
		Camelina	283 ^C^	442 ^B^	496 ^bA^	524 ^aA^	126
ADF	g kg^−1^ DM	Amaranth	246 ^B^	321 ^A^	326 ^A^	349 ^A^	524
		Borage	180 ^B^	282 ^A^	283 ^bA^	304 ^aA^	46
		Camelina	259 ^C^	369 ^B^	408 ^bA^	434 ^aA^	94
EE	g kg^−1^ DM	Amaranth	15	13	14	14	8.1
		Borage	26 ^aA^	20 ^bAB^	19 ^B^	17 ^B^	2.1
		Camelina	34 ^A^	27 ^B^	22 ^C^	22 ^C^	2.2
ADMD_AD_^II^	g kg^−1^ DM	Amaranth	968 ^A^	869 ^aB^	846 ^abB^	781 ^bB^	793
		Borage	941 ^aA^	907 ^bAB^	902 ^bAB^	887 ^B^	51
		Camelina	914 ^A^	778 ^B^	720 ^B^	651 ^C^	169
NDFD_AD_^II^	g kg^−1^ NDF	Amaranth	912	697	606	575	3032
		Borage	800 ^a^	713 ^ab^	693 ^b^	671 ^b^	488
		Camelina	695 ^A^	497 ^B^	436 ^B^	334 ^C^	731

^a–c^ = *p* < 0.01; ^A–D^ = *p* < 0.01 on the same row. Abbreviations: DM = dry matter; CP = crude protein; NDF = neutral digestible fiber; ADF = acid digestible fiber; EE = ethereal extract; ADMD_AD_^II^ = dry matter digestibility measured using Ankom Daisy^II^; NDFD_AD_^II^ = neutral detergent fiber digestibility measured using Ankom Daisy.

**Table 4 animals-12-02320-t004:** Lsmean and mean square of error (MSE) of the calculated parameters at the vegetative, elongation, reproductive, and seed stages (Amaranth DFE = 10; Borage DFE = 6; Camelina DFE = 11), when using the formulae for legumes (_lg_) and grasses (_gr_).

Parameter	Unit	Species	Stage of Maturity				MSE
			Vegetative	Elongation	Reproductive	Seed	
NFC	g kg^−1^ DM	Amaranth	228 ^b^	279 ^ab^	338 ^a^	212 ^b^	1945
		Borage	227 ^b^	253 ^ab^	305 ^a^	319 ^a^	613
		Camelina	324 ^A^	279 ^B^	272 ^B^	287 B	145
FA	g kg^−1^ DM	Amaranth	4.7	2.5	3.9	4.2	8.1
		Borage	16 ^aA^	10 ^bAB^	9 ^B^	7 ^B^	2.1
		Camelina	24 ^A^	17 ^B^	12 ^C^	12 ^C^	2.2
NDFn	g kg^−1^ DM	Amaranth	335 ^cB^	413 ^bA^	397 ^bAB^	479 ^aA^	668
		Borage	268 ^C^	302 ^bAB^	297 ^B^	320 ^aA^	15
		Camelina	263 ^C^	411 ^B^	462 ^bA^	487 ^aA^	109
ADMD_ADF_	g kg^−1^ DM	Amaranth	698 ^A^	639 ^B^	635 ^B^	617 ^B^	318
		Borage	749 ^A^	669 ^aB^	668 ^aB^	652 ^bB^	28
		Camelina	687 ^A^	602 ^B^	571 ^aC^	551 ^bC^	57
DMI_NDF_	%	Amaranth	3.4 ^aA^	2.7 ^B^	2.8 ^bB^	2.3 ^B^	0.053
		Borage	4.2 ^A^	3.7 ^aBC^	3.8 ^B^	3.5 ^bC^	0.0029
		Camelina	4.2 ^A^	2.7 ^B^	2.4 ^C^	2.3 ^C^	0.0065
DMI_lg_	%	Amaranth	4.6 ^A^	3.4 ^B^	3.2 ^B^	2.7 ^B^	0.097
		Borage	5.1 ^A^	4.4 ^aB^	4.4 ^aB^	4.1 ^bB^	0.0054
		Camelina	4.9 ^A^	2.8 ^B^	2.4 ^C^	2.0 ^D^	0.0083
DMI_gr_	%	Amaranth	2..4	2.5	2.2	2.3	0.092
		Borage	2.4	2.5	2.6	2.6	0.0059
		Camelina	2.6 ^A^	2.5 ^aAB^	2.3 ^bBC^	2.1 ^cC^	0.0074
TDN_lg_	g kg^−1^ DM	Amaranth	632 ^A^	584 ^AB^	574 ^AB^	514 ^B^	903
		Borage	572	548	578	582	486
		Camelina	675 ^A^	572 ^B^	532 ^C^	488 ^D^	81
TDN_gr_	g kg^−1^ DM	Amaranth	575 ^a^	552 ^ab^	553 ^ab^	500 ^b^	628
		Borage	530	514	546	553	444
		Camelina	643 ^A^	567 ^B^	539 ^C^	514 ^D^	43
NE_L_ADF_		Amaranth	1.5 ^A^	1.4 ^B^	1.4 ^B^	1.3 ^B^	0.0015
		Borage	1.6 ^A^	1.4 ^B^	1.4 ^B^	1.4 ^B^	0.0001
		Camelina	1.47 ^A^	1.28 ^B^	1.21 ^aC^	1.17 ^bC^	0.0003

^a–c^, = *p* < 0.05; ^A–D^ = *p* < 0.01 on the same row. Abbreviations: NFC = non-fibrous carbohydrate; FA = fatty acids; NDFn = nitrogen-free neutral detergent fiber; ADMD_ADF_ = dry matter digestibility ADF based; DMI_NDF_ = dry matter intake NDF based; DMI_lg_ = dry matter intake for legumes; TDN_lg_ = total digestible nutrient for legumes; TDNgr = total digestible nutrient for grasses; NE_L-ADF_ = net energy of lactation (ADF based).

**Table 5 animals-12-02320-t005:** Lsmean and mean square of error (MSE) of the calculated indexes at the vegetative, elongation, reproductive, and seed stages (g kg^−1^ DM; Amaranth DFE = 10; Borage DFE = 6; Camelina DFE = 11), when using the formulae for legumes (_lg_) and grasses (_gr_).

Species	Index	Stage of Maturity				MSE
		Vegetative	Elongation	Reproductive	Seed	
Amaranth	RFV_ADF_	182 ^aA^	134 ^B^	138 ^bB^	112 ^B^	235
	RFV_vt_	252 ^A^	182 ^B^	184 ^B^	141 ^B^	396
	RFQ_lg_	238 ^Aa^	162 ^aB^	151 ^abB^	112 ^bB^	309
	RFQ_gr_	115	110	99	93	327
Borage	RFV_ADF_	242 ^B^	192 ^B^	194 ^B^	176 ^C^	8.2
	RFV_vt_	299 ^A^	260 ^aBC^	262 ^B^	240 ^bC^	14
	RFQ_lg_	235	197	208	194	101
	RFQ_gr_	104	104	116	116	62
Camelina	RFV_ADF_	226 ^A^	127 ^B^	107 ^C^	98 ^C^	25
	RFV_vt_	301 ^A^	164 ^B^	135 ^C^	116 ^D^	36
	RFQ_lg_	270 ^A^	133 ^B^	103 ^C^	78 ^D^	29
	RFQ_gr_	136 ^A^	114 ^aB^	99 ^bBC^	87 ^cC^	24

^a, b, c^ = *p* < 0.05; ^A, B, C, D^ = *p* < 0.01 on the same row. Abbreviations: RFV = relative feed value; RFV_ADF_ = relative feed value ADF based; RFV_vt_ = relative feed value measured in vitro; RFQ = relative forage quality; RFQ_lg_ = relative forage quality for legumes; RFQ_gr_ = relative forage quality for grasses.

## Data Availability

Not applicable.

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
