# Peer review of "Laboratory Analyses Used to Define the Nutritional Parameters and Quality Indexes of Some Unusual Forages"

_animals, 2022, doi:10.3390/ani12182320_

Round 1
Reviewer 1 Report
Line 268 to 269: ”Amaranth, borage and camelina showed a good quality, in particular at vegetative stage, that decreased during growth, maintained high values up to seed stage”. In the discussion of the data and conclusions sessions, the authors affirm that the quality of the unusual forages plants was adequately evaluated through to the traditional analyzes applied in the forage evaluation. However, we suggest that the authors consider the possible effects of secondary compounds, such as essential oils, tannins, saponins that could be present in these plants, which can influence the intake and digestibility of these unusual forage. Anti-qualitative factors have a pronounced effect on intake of forage plants and should be discussed when evaluating unusual plants. We suggest approaching these topics in the article, including specific data related to the presence or absence of anti-quantitative factors in the evaluated unusual plants.

Reviewer 2 Report
Comments to Authors: MDPI animals-1869459, Laboratory analyses to define nutritional parameters and quality indexes in unusual forages
The manuscript provides potentially valuable information for scientists and end users regarding the use of various already-developed forage quality indicators to evaluate and compare selected species not originally included in the equation development.
First, after the few other deficiencies, are addressed, the manuscript should be reviewed by someone who’s native language is English as there are numerous grammatical errors and wording used that does not convey a clear point. A few examples are highlighted below.
Title and throughout: “Indexes” and “indices” are both valid as the plural form of index and both are used in the manuscript. Except for reference titles, which cannot be changed, I recommend using consistent terminology throughout the manuscript.
Simple Summary and Abstract:
Lines 10 & 22: “Relative Forage Quality”
Line 13: “production”
Line 15: “. . . to evaluate and rank forages . . .”
Line 16: “differentiates”
Line 20: “value”
Lines 22-23: Define RFQ parenthetically; “. . . legume, compared to the grass formula would seem . . .”
Line 23-24: “. . . unusual forages as RFQ revealed . . .” Avoid using pronouns that refer back to a previous sentence.
Introduction:
Line 35: “Forage quality is defined . . . , measured in the laboratory . . .”
Line 40: “Over the course of time, several . . .”
That should be enough examples of grammatical errors or wording that needs clarified. There are many more throughout the manuscript that will likely only be resolved by a reviewed and corrected by someone whose native language is English.
Line 72: Introduce how amaranth, borage, and camelina are being used in livestock diets, or can be. Why did you select those species for the study? Also, it would be good to globalize need for the study and the results. Perhaps other non-legume and non-grass species have been similarly evaluated and mentioning those as also being unusual forages would help broaden the scope of application.
Materials and Methods:
Line 89: “Fairport, NY USA” When countries have states or provinces, it is important to include those in the location description, as there are many cities with the same name, especially throughout the USA in different states. When the city, state/province country format is used, no comma is used between the state/province and nation. A comma always follows the city name when either a state/province or nation name is provided. Check line 118.
Line 101 and elsewhere: Either non-fibrous or nonfibrous; never “non fibrous”
Line 116: “Data were grouped into four . . . “
Line 118: “. . . [27] to compare biological stages within each species. Species differences were not analyzed.” I assume that you did not run repeated measurements analysis in GLM to compare linear, quadratic, and cubic effects to compare species across biological stages. That would have really increased the power of the study to evaluating the forage quality indices/indexes AND to help growers determine the optimum harvest maturity to maximize nutritive value and DM yield of the species studied. Regarding “biological stages (originally used at line 116),” “phenological stage” is used at line 125 and “development stage” is used at line 212. Be consistent in terminology. Since these are all variations of “stage of maturity” and that is more widely recognized, I recommend using that throughout the manuscript to describe the sampling stages.
Please describe the replication of each species used in the analysis. Each study should have been replicated and may have multiple years. It would be appropriate to know how much replication supports the results. More on that.
Table 2: Each formula begins with the result (e.g., FA = ), except the formulae for NFC and DMDADF. Be consistent. Also, NDF, and subscripts lg, gr, and vt are not defined in the footnote. DMDvt is defined in the table for that formula, but not RFVvt. In the table footnote, clarify “%WB=percentage of weight basis.” Should that be % of DM? Otherwise, I could not find that term used in the table. If it is not used, delete the definition.
Results:
Line 124: There is no such word as “specie” The term “species” is both singular and plural.
Line 127 and elsewhere in Results and Discussion sections: Do not repeat data from tables or figures in the text. Use table/figure callouts early and often to refer readers to those places to find the data.
Table 3: Title: Add a sentence indicating the level of replication for each species if different. Otherwise, it could be something like: “Values are the means of x replicates and y years (studies/locations if repeated within the same year) for each species.” Should all data, except RFV and RFQ be reported in SI units (g kg-1 DM, except for NDFD, which would be g kg-1 NDF)? Footnote: Delete “; A, B, C, = P<0.01” because no capital letters are used. A comparison of species across sampling stages using repeated measurements could really enhance the information gained and strengthen the comparisons of the various indexes/indices. Here are some sample codes for that in SAS:
PROC GLM;
CLASS SPECIES REP(SPECIES);
MODEL DM1-DM4=REP(SPECIES) SPECIES;
TEST H=SPECIES E=REP(SPECIES);
REPEATED STAGE (1 2 3 4)POLYNOMIAL/PRINTE NOM SUMMARY HTYPE=1 ETYPE=1;
You could use this analysis for all variables.
Table 4: First, move to come after the first callout at current line 156. Second, the DMI and TDN sections are confusing. Second, add a statement about replication. Third, I recommend deleting DMI and TDN and left aligning DMIlg, DMIgr, TDNlg, and TDNgr to be aligned with the other variables. Should data be in SI units rather than %?
Line 164: “The DMINDF was . . .” This term has been defined. No need to define again in the text.
Table 5: See comment about left aligning variables in Table 4.
Table 5 and Figure 1 report the same data. That is redundant; choose one or the other. Both have value. Differences within variables over sampling dates for each species in the figure can be indicated by assigning error bars an equal value to the Tukey HSD, whereas, the letters in Table 5 accomplish that. Figure 1 lends itself very well to reporting the repeated measurements analysis, which also will provide an LSD within each sampling stage to compare species. You might be able to assign an HSD test, but I am not as familiar with that for the repeated measurements procedure in SAS. Then you could assign letters or an error bar within each sampling date.
Alternatively, if you can determine how to arrange the data (probably as a single column with index as a variable), you might be able to compare indexes within species and maybe even across species by adding index and species x index to the codes shown above. If repeated measures is done, you can place the p-values for the regression component on the pane to show how the indexes changed relatively over time.
At any rate, for Figure 1, I recommend increasing the size of the lines and markers for each data series to make the figure easier to read. Also, the abbreviation for each index should be defined in the caption. Finally, you can widen the panes and use the name of the growth stage as the x-axis label instead of numbers that had to be defined. The way that was done decreases the figure quality.
Discussion: Remember, when referring to your results, provide a table or figure callout.
Lines 239-240: Species comparisons are not reported. Hence, unless you describe a statistical comparison of species (along with species x sampling stage interaction or repeated measurements), you have no bases to support this statement.
Lines 261-263: No statistics are shown on Figure 1; therefore, you cannot state that changes in nutritional value in maturity (there is the appropriate term for sampling stages) was detectable. Table 5 shows that, though. So, you could refer to Table 5, but, again, you do not need both Table 5 and Figure 1 and one of those presentations merely takes up valuable space online.
Conclusions:
Lines 268-269: “. . . in particular, at the vegetative stage, that decreased over time, but still maintained . . .” (One last reminder to have the revised manuscript reviewed someone whose native language is English to correct grammar and clarify statements, maintaining your intent.)
References appear to be appropriately formatted.
The information developed during this study has potential to be of great value for the scientific community, as well as end users. Please follow the recommendations of the reviewers that are supported by the editors to revise and resubmit this manuscript for publication. I look forward to the revision!
Round 2
Reviewer 2 Report
Thank you for completing an excellent revision. I only have a few minor changes, two of which were caused trying to address one of my comments.
Lines 99 & 130: While we would say or write something about "the USA", when it comes to using it in an address, the article (the) would not be used. Hence, at line 99, it should be "Fairport, NY USA" and at line 130, it should be "Cary, NC USA" with no comma between the state (NY or NC) and USA (the country). Sorry for that confusion.
Lines 122 & 124: Since RFV and RFQ are previously defined in the main text at lines 45-46 & 68, respectively and neither abbreviation is used to start the sentence, the full term need not be used. Therefore, line 122 should start "The RFV index . . ." and line 124 should start "The RFQ index . . ."
Otherwise, adding the text to the conclusion about why the species were selected addresses that question and regarding the statistical analysis and presentation of the data, showing the df in table titles addresses that concern very well.
Thank you also for the kind comment regarding the review. My philosophy is that we should help others be successful in research and publication and in the process, we also learn.
